# Calcium Delivery by Electroporation Induces In Vitro Cell Death through Mitochondrial Dysfunction without DNA Damages

**DOI:** 10.3390/cancers12020425

**Published:** 2020-02-12

**Authors:** Laure Gibot, Audrey Montigny, Houda Baaziz, Isabelle Fourquaux, Marc Audebert, Marie-Pierre Rols

**Affiliations:** 1Institut de Pharmacologie et de Biologie Structurale, Université de Toulouse, CNRS, UPS, 31077 Toulouse, France; gibot@ipbs.fr (L.G.); montigny@ipbs.fr (A.M.); baaziz@ipbs.fr (H.B.); 2Centre de Microscopie Électronique Appliquée à la Biologie, CMEAB, 133 route de Narbonne, 31062 Toulouse CEDEX, France; isabelle.fourquaux@univ-tlse3.fr; 3Toxalim, Université de Toulouse, INRAE-UMR1331, ENVT, INP-Purpan, UPS, 31027 Toulouse, France

**Keywords:** calcium electroporation, electrochemotherapy, spheroids, magnesium, genotoxicity, mitochondria

## Abstract

Adolescent cancer survivors present increased risks of developing secondary malignancies due to cancer therapy. Electrochemotherapy is a promising anti-cancer approach that potentiates the cytotoxic effect of drugs by application of external electric field pulses. Clinicians proposed to associate electroporation and calcium. The current study aims to unravel the toxic mechanisms of calcium electroporation, in particular if calcium presents a genotoxic profile and if its cytotoxicity comes from the ion itself or from osmotic stress. Human dermal fibroblasts and colorectal HCT-116 cell line were treated by electrochemotherapy using bleomycin, cisplatin, calcium, or magnesium. Genotoxicity, cytotoxicity, mitochondrial membrane potential, ATP content, and caspases activities were assessed in cells grown on monolayers and tumor growth was assayed in tumor spheroids. Results in monolayers show that unlike cisplatin and bleomycin, calcium electroporation induces cell death without genotoxicity induction. Its cytotoxicity correlates with a dramatic fall in mitochondrial membrane potential and ATP depletion. Opposite of magnesium, over seven days of calcium electroporation led to spheroid tumor growth regression. As non-genotoxic, calcium has a better safety profile than conventional anticancer drugs. Calcium is already authorized by different health authorities worldwide. Therefore, calcium electroporation should be a cancer treatment of choice due to the reduced potential of secondary malignancies.

## 1. Introduction

An alternative approach to classical chemotherapy, electrochemotherapy (ECT), has developed in recent decades and is currently used in more than 150 centers and clinics throughout Europe [1]. Electrochemotherapy treatment was standardized in the framework of the European Standard Operating Procedure on Electrochemotherapy (ESOPE) multicenter trial, first released in 2006 and recently updated [2]. It consists of associating cytotoxic drug injection with the application of calibrated electric field pulses delivered locally at the tumor site. Indeed, in the 1970s, it was demonstrated that the application of calibrated external electric field pulses can transiently permeabilize the vesicular cell membrane [3]. Further developments of this approach showed that transient permeabilization of the plasma membrane allows non-permeant or low permeant cytotoxic drugs to enter the cytoplasm in large quantities and thus potentiate its activity. The proof of concept was led with bleomycin, a non-permeant molecule with a very high molecular weight and intrinsic cytotoxicity which was dramatically enhanced by 700-fold after electroporation, compared to the drug alone [4]. On the contrary, only a 1.1–1.3-fold increase of cytotoxicity was observed with lipophilic drugs able to rapidly diffuse through the plasma membrane, such as vinblastine [5]. This enhanced cytoplasmic penetration, spatially and temporarily localized in the tissue between the electrodes diminishes the quantity of injected antitumor drugs, thus limiting short-term and long-term side effects. Nowadays, electrochemotherapy is associated with the application of external electric field pulses and injection of hydrophilic drugs such as bleomycin and cisplatin. A study comparing the efficiency of electrochemotherapy of cisplatin with intratumoral administration of cisplatin alone on cutaneous tumor lesions in breast cancer indicated that ECT displayed a better complete objective response to treatment than cisplatin alone (100% and 83%, respectively), with 33% complete response for ECT and 0% for cisplatin alone [6]. Interestingly, according to a recent meta-analysis of all available clinical data on electrochemotherapy, the objective treatment response of evaluated studies was 83.91% (95% CI: 79.15–88.17%) for bleomycin and 80.82% (95% CI: 66.00–92.36%) for cisplatin [7] with a good preservation of healthy tissues. Long-term effects of electrochemotherapy are still to be determined, as the potential occurrence of secondary malignancies has never been reported until now, after more than a decade of use.

During the last years, calcium electroporation was proposed as a potential novel anticancer treatment, where high concentrations of calcium (CaCl_2_) are introduced in cell cytoplasm thanks to electroporation, inducing cell necrosis. This innovative approach already demonstrated its efficiency in vitro on two-dimensional (2D) cell monolayer, in 3D spheroid tumor model, in vivo, and in human clinical studies [8,9,10,11,12]. This treatment modality is fairly easy to implement. Calcium is injected locally into the tumors and electric pulses are administered immediately thereafter. Recently, a randomized double-blinded phase II study compared calcium electroporation with bleomycin electrochemotherapy against cutaneous metastasis [11]. There was no statistically significant difference between the two treatments. Remarkably, calcium electroporation presented fewer secondary effects than electrochemotherapy (itching, exudation, hyperpigmentation). Calcium electroporation was also shown as a safe treatment on mucosal head and neck cancers with no signs of hypercalcemia, cardiac arrhythmias, or severe adverse events [12]. Perspectives for calcium electroporation are therefore highly promising.

Based on that, our study proposes to answer two specific questions in order to better understand cellular toxic mechanisms induced by calcium electroporation and keep promoting its use as a safe, efficient, and innovative antitumor treatment. First, does calcium electroporation treatment, in addition to being cytotoxic, present a genotoxic profile like other classical antitumor drugs such as cisplatin or bleomycin? Bleomycin as well as cisplatin are used to set up the genotoxic assay and validate the proven effectiveness of electroporation with these two drugs. Second, is the cytotoxic effect of calcium electroporation specifically due to calcium ion or to the osmolarity of the solution? Magnesium (MgCl_2_) solutions at the same osmolarity as calcium (CaCl_2_) solutions were used to treat the cells and find responses.

To answer these questions, we used human normal primary dermal fibroblasts and human colorectal cancer cell line HCT-116. The first part of the study was led on cell cultures grown in 2D monolayers. The other part was led on a 3D tumor model named spheroid, particularly suitable for in vitro study of electroporation [13,14,15] and closer to in vivo reality than monolayers [16].

## 2. Results

### 2.1. Determination of Optimal Electrical Parameters to Induce Reversible Cell Electropermeabilization

Typical electrical parameters used in electrochemotherapy are eight pulses lasting 100 µs, applied at 1 Hz frequency [17]. The electric field intensity to apply to efficiently induce reversible plasma membrane permeabilization without affecting cell viability depends on the cell type and cell organization (in 2D or in 3D models) and therefore has to be determined before the experiment. For normal dermal fibroblasts and tumor HCT-116 cells grown in monolayers, a 500 V/cm intensity appeared to be the optimal value (Figure 1).

In this condition, more than 90% of cells were efficiently electropermeabilized, as determined by propidium iodide uptake, while their viability was unaffected. In 3D colorectal tumor spheroids, cells display a distinct shape and size than in monolayers and therefore require different electric field intensity. We took advantage of our previous study and choose 1000 V/cm as the optimum intensity allowing efficient electropermeabilization while ensuring cell viability [9]. 

### 2.2. Electrochemotherapy Potentiates Genotoxic and Cytotoxic Effects of Cisplatin and Bleomycin, both in Normal and Tumor Cells

Whether for tumor HCT-116 cells or for normal dermal fibroblasts, cisplatin and bleomycin were genotoxic as revealed by induction of the phosphorylation of histone H2AX, a biomarker of global DNA damage [18]. As shown in Figure 2, compared to the control, a significant increase in genotoxicity appeared above 10 µM for cisplatin alone and 50 nM for bleomycin alone in a concentration-dependent manner of the antitumor drugs cisplatin (1, 10, 50, 100 µM) and bleomycin (5, 50, 500, 1000 nM). When associated with electroporation, the genotoxicity of cisplatin tended to increase but statistical significance was not observed. In this experimental condition, a significant decrease in the viability of cancer cells was observed, while this effect was not statistically significant for normal dermal fibroblasts. Concerning bleomycin, it is obvious that the association with electroporation hugely potentiated both its genotoxic and cytotoxic effects. Indeed, even the lowest concentration of bleomycin (i.e., 5 nM) induced genotoxicity in HCT-116 cells (1.6-fold induction of γH2AX compared to control condition), but when associated with electroporation, it raised to 7.5-fold. Similarly, for 50 nM of bleomycin in normal fibroblasts, fold induction of γH2AX switched from 2 when incubated alone to 14 when associated with electroporation. Increasing genotoxic effect was also correlated with a higher cytotoxic effect. When relative cell count was below 50% of the control condition, no genotoxicity was presented on the graphics to avoid false positive genotoxic results due to apoptosis [19].

### 2.3. Calcium Electroporation Induces Cytotoxicity without any Genotoxicity, while Magnesium Electroporation Has No Toxic Effect

Whether in tumor HCT-116 cells or in normal dermal fibroblasts, neither calcium nor magnesium (1, 5, or 10 mM) associated with electroporation induced genotoxicity (Figure 3). When associated with electroporation, 5 and 10 mM calcium induced a statistical significantly reduced cell viability, respectively to 83% and 86% viability in HCT-116, and 13% and 5% viability in dermal fibroblasts. Interestingly, while CaCl_2_ and MgCl_2_ solutions present the same osmolarity, no significant toxic effect of magnesium was observed when associated with electroporation.

### 2.4. Calcium Electroporation Induces ATP Depletion and Mitochondrial Membrane Depolarization

We analyzed the intracellular ATP content in short time post-treatment (i.e., in 5 min) (Figure 4) as ATP is widely accepted as a reliable and valid marker of cell viability [20]. ATP leakage is associated with plasma membrane electropermeabilization. Therefore, we expected to observe a fall of the intracellular ATP following electroporation alone, but the decrease was not statistically significant 5 min after application of the electric field, indicating an efficient resealing under these electrical conditions; this is in agreement with the fact that under these electrical conditions cells are efficiently permeabilized while their viability remains unaffected (Figure 1). Contrariwise, when electroporation is associated with 5 mM calcium or magnesium, a statistically significant drop in intracellular ATP content was quantified. In HCT-116 cells the decrease of ATP content was around 25% and reached 60% in normal cells.

Observation of mitochondria’s ultrastructure indicates that calcium associated to electroporation induced mitochondria swelling and disorganized mitochondrial inner membranes while magnesium associated to electroporation modified the contrast of mitochondria (Figure 5).

We assessed the mitochondrial membrane potential using a fluorescent probe that rapidly accumulates in normal mitochondria (Figure 6). Staining is dependent on mitochondrial membrane potential and is lost when mitochondria become depolarized. A dramatic event leading to mitochondrial membrane depolarization in both normal and tumor cells occurred within 5 min following cell electroporation with calcium, which was not the case with magnesium. 

### 2.5. Apoptosis Pathway Is Not Induced by Mitochondrial Perturbation Rapidly Observed after Calcium Electroporation 

Calcium associated to electroporation displayed a cytotoxic effect as well as a short-term alteration of mitochondria architecture and function that could lead to cell death. For that purpose, apoptosis was assessed in general and more specifically through the mitochondrial pathway (involving caspase-9). Caspase-9, an initiator caspase, is a key player in the mitochondrial pathway of apoptosis initiated in response to agents or insults that trigger, for example, the release of cytochrome c from mitochondria [21]. Caspase-9 and caspases-3/7 activities were quantified over 2 h post treatment with 5 mM CaCl_2_ associated with electroporation (Figure 7). No specific activity of these caspases was observed during this time. Furthermore, despite the rapid and massive alteration of mitochondrial membrane potential within the first minutes following calcium electroporation, no release of cytochrome c was observed after 5 min (Figure 8), which was consistent with the non-activation of caspase-9 over 2 h following the treatment.

### 2.6. While Calcium Electroporation Leads to Sustainable Cell Death in Tumor Spheroids, Magnesium Electroporation Solely Temporarily Disaggregates It

We took advantage of the ability of HCT-116 tumor cells to produce homogeneous spheroids to study the response of this 3D tumor tissue model to calcium or magnesium electroporation. Thus, we observed the long-term macroscopic aspect of tumor spheroids to CaCl_2_ and MgCl_2_ treatments (Appendix A). The first observation is that immediately after treatment (the first sequence of the movie) spheroids electroporated with calcium or magnesium swelled and presented blurry margins, contrary to other experimental conditions where spheroids were tightly packed. Spheroids of the control condition without electroporation, 168 mM CaCl_2_ and 168 mM MgCl_2_ conditions kept growing normally over seven days. The spheroids exposed to electroporation alone presented a macroscopic cohesive aspect but grew slower than the control conditions cited above. After the first culture medium changing (18 h post treatment), a massive detachment of single cells and clusters of cells from the surface of the spheroids was observed in calcium and magnesium electroporation conditions. Interestingly, two distinct behaviors were observed. The vast majority of detached cells from spheroids treated with calcium electroporation remained inactive. Contrariwise, cellular clusters detached from spheroids treated with magnesium electroporation kept proliferating and moving towards each other to finally fuse together with the original remaining core of the spheroid, forming a giant spheroid. After seven days of experiments, spheroids electroporated with calcium presented a smaller size than all other conditions, contrary to those treated with magnesium associated with electroporation which kept growing. These distinct behaviors are linked to the results obtained on monolayer where calcium electroporation led to high cytotoxicity while magnesium electroporation had no effect on cell viability.

## 3. Discussion

Even if calcium and magnesium solutions used to treat 2D cells or 3D tumor spheroids presented the same osmolarity, cellular effects at the short-term and long-term were not comparable. These observations underline the major importance of the nature of the ion used to treat cells by electroporation. It was previously reported in vitro that at equimolar concentrations, both calcium chloride and calcium glubionate presented the same effects on cytotoxicity when associated with electroporation [22]. Calcium, as a universal carrier of biological information, plays a pivotal role in cell signaling and homeostasis [23]. Electroporation associated to calcium induces a massive influx of this ion within cells, disrupting the intracellular calcium homeostasis. Mitochondrial calcium concentration was shown to regulate respiration, ATP, and reactive oxygen species (ROS) production [24]. These authors demonstrated that micromolar calcium concentrations inhibit the capacity of mitochondria to utilize membrane potential for ATP production. Furthermore, it is known that when intracellular calcium dramatically increases, as is obviously the case after calcium electroporation, Ca2+-ATPases rapidly activate and consume the intracellular pool of ATP [25]. This bundle of evidence could explain the massive intracellular ATP depletion observed in our study within minutes after calcium electroporation and that previously reported in the literature on calcium electroporation [8,26,27]. 

More and more scientific evidences underline the link between ATP production and apoptosis in mitochondria, mediated by its ultrastructure remodeling [28]. In normal conditions, cells are able to maintain stable levels of ATP and mitochondrial membrane potential. However, a long-lasting drop or rise of mitochondrial membrane potential versus normal levels may induce unwanted loss of cell viability [29]. We may assume that the brutal and drastic drop of mitochondrial membrane potential induced by calcium electroporation initiates cellular processes leading to cell death, explaining the high cytotoxic potential of this treatment. In our experimental set-up, we gave evidence that cell death induced by calcium electroporation was not driven by the apoptosis pathway. According to previous studies on calcium electroporation, cellular mechanisms underlying cytotoxic effects of calcium electroporation lay on intracellular ATP depletion and necrosis induction [8,26,27,30]. 

It has to be underlined that different selectivity to calcium electroporation between normal and cancer cells was observed both in 3D tumor spheroids, in vivo and in clinical studies [30,31,32], while no selectivity of calcium electroporation for cancer cells compared to normal cells was observed in the experimental conditions of our study performed in vitro in 2D, where the external volume of calcium is very large compared to the intracellular volume, as already reported in most previous studies [27,33]. All biological observations arising from cell treatment with calcium electroporation (cytotoxicity, ATP intracellular depletion, drop of mitochondrial membrane potential, etc.) were indeed also observed in tumor HCT-116 cells and normal dermal fibroblasts monolayers. Further experiments on distinct tumor cells lines and normal cell types should be led to determine the specificity of calcium electroporation.

Histone H2AX plays a crucial role as a platform on which DNA repair complexes are formed at the sites of DNA damage [34]. Thus, through the in situ detection of its phosphorylated form, γH2AX was proposed as a promising tool in evaluating the genotoxic potential of chemicals [19]. In normal conditions, water-soluble antibiotic bleomycin solely crosses the plasma membrane through ligand-receptor binding process [35]. However, when associated with electroporation, millions of bleomycin molecules are internalized by cells, leading to a very rapid (within seconds) DNA fragmentation into oligonucleosomal-sized fragments [36]. Depending on the ratio of single strand breaks and double strand breaks induced by bleomycin, cells die by apoptosis, pseudo-apoptosis, or mitotic cell death [37]. In this study, the γH2AX assay confirmed that cisplatin and bleomycin presented genotoxic and cytotoxic properties, especially when associated with electroporation. It also allowed for quantification of the potentiating effect of electroporation to deliver these drugs. Thus, cisplatin electroporation displayed low advantage contrary to cisplatin alone: one-fold increase of genotoxicity for both HCT-116 and normal dermal fibroblasts whatever the concentration applied. Oppositely, bleomycin electroporation led to 7- and up to 10-fold genotoxicity increase respectively in HCT-116 and fibroblasts compared to bleomycin alone. Thus, electrochemotherapy uses less drugs than conventional chemotherapies to treat cancers with the same efficiency. This difference between compounds may be linked to their physical properties with cisplatin being more effective through diffusion into cells. 

In parallel to this observation for known genotoxic agents, this validated method demonstrates for the first time that calcium electroporation displayed an important cytotoxic effect, without generating any genotoxicity. Thanks to these particular characteristics, electroporation associated to calcium could become the gold standard of inexpensive and simple antitumor therapy. Indeed, it is important to keep in mind that with the current anticancer treatments there is more and more remission, but survivors present increased risks of developing secondary malignancies due to the genotoxic effects of the proposed conventional therapies [38]. Thus, a meta-analysis performed on 2,116,163 patients in the USA suffering from the 10 most common cancer sites pointed out that approximately 1 in 12 cancer survivors (8%) develops a second primary malignancy at some point [39].

## 4. Materials and Methods 

### 4.1. Chemicals

Bleomycin sulfate (Bleo) was purchased from Merck-Millipore (Molsheim, France). PrestoBlue reagent was purchased from Invitrogen Life Technologies (Saint Aubin, France). Cis-diamminedichloroplatinum (II) (Cisplatin, Cispt), calcium chloride (CaCl_2_), magnesium chloride (MgCl_2_), etoposide (VP16), staurosporine, and propidium iodide (PI) were purchased from Sigma-Aldrich (St Quentin Fallavier, France).

### 4.2. Cell Culture

Two distinct types of human cells were used. Colorectal tumor cell line HCT-116 was bought from ATCC (#CCL-247) and primary normal dermal fibroblasts were isolated from a 3-year-old foreskin as previously described [40,41]. Both cell types were grown in Dulbecco’s modified Eagles medium (Gibco-Invitrogen, Carlsbad, CA, USA) containing Glutamax, 4.5 g/L glucose and pyruvate, supplemented with 10% (v/v) heat inactivated fetal calf serum, 100 U/mL penicillin, and 100 μg/mL streptomycin. All along the experiments, cells tested negative for mycoplasma (MycoAlert mycoplasma detection kit, Lonza, Walkersville, MD, USA). Cell cultures were maintained in a humidified atmosphere at 37 °C containing 5% CO_2_.

### 4.3. Tumor Spheroid Production

Three-dimensional (3D) tumor spheroids were produced thanks to the non-adherent technique as previously described [9,42]. Briefly 5000 HCT-116 cells were seeded in ultra-low attachment 96-well plates (Corning, Fisher Scientific, Loughborough, UK). Spheroids were grown for 5 days at 37 °C in a humidified atmosphere containing 5% CO_2_ before treatment.

### 4.4. Electroporation of 2D Monolayers and 3D Tumor Spheroids

Bleomycin and cisplatin drugs were diluted in pulsing buffer (10 mM K_2_HPO_4_/KH_2_PO_4_, 250 mM sucrose, and 1 mM MgCl_2_ in sterile water, pH 7.4) [14] while, in order to avoid precipitation, CaCl_2_ and MgCl_2_ were diluted in HEPES buffer (10 mM HEPES, 250 mM sucrose, and 1 mM MgCl_2_ in sterile water, pH 7.4) as previously described [8]. For 2D experiments, normal dermal fibroblasts and tumor HCT-116 cells were grown as monolayer in 96-well plates. After washing with phosphate-buffered saline solution (PBS), cells were covered with 50 µL of pulsing or HEPES buffer containing different concentrations of drugs and incubated for 5 min at room temperature. Two stainless, flat, parallel electrodes (0.35 cm between electrodes) designed specifically to 96-well plate geometry (Megastil, Ljubljana, Slovenia) were applied to the bottom of the well. Defined electric field (8 square-wave pulses of 100 μs, 1 Hz, and 500 V/cm) was delivered at room temperature by Electro cell S20 generator (LeroyBiotech, Saint Orens de Gameville, France). Special attention was paid to keep constant the time of contact with cytotoxic drugs (bleomycin, cisplatin) or salts (CaCl_2_ or MgCl_2_), so that it was the same with and without electroporation (i.e., 6 min). After treatment, cells were washed once with 300 µL of PBS before addition of 100 µL of cell culture medium and placed in a humidified atmosphere at 37 °C containing 5% CO_2_ until analysis. The concentrations (1, 5, 10 mM) of calcium, and consequently those of magnesium, chosen to treat the cells grown in monolayers were similar to those used in previously published in vitro studies [8,22]. For 3D tumor spheroids experiments, a single spheroid was placed between two stainless steel, flat, parallel electrodes (0.4 cm between electrodes) in 100 μL of HEPES buffer alone or HEPES buffer containing 168 mM CaCl_2_ or 168 mM MgCl_2_. This high calcium concentration was previously chosen for in vivo experiments [8,33] because it corresponds to the concentration of CaCl_2_ solutions commercially available in hospitals. In addition, when associated to electroporation, 168 mM CaCl_2_ previously demonstrated efficacy against 3D tumor spheroids in vitro [9]. After 5 min of incubation with CaCl_2_ or MgCl_2_ at room temperature, defined electric field allowing transient cell electropermeabilization (8 square-wave pulses of 100 μs, 1 Hz, and 1000 V/cm) was delivered at room temperature by Electro cell S20 generator as previously described [9]; see Figure A1 for the definition of electric parameters. After treatment, spheroids were washed twice with PBS before being placed in 300 µL of cell culture medium in a 96-well plate and further placed in a humidified atmosphere at 37 °C containing 5% CO_2_ until analysis.

### 4.5. Determination of Cell Electropermeabilization and Viability after Electroporation

To detect cell electropermeabilization, adherent cells grown in monolayer were submitted to electroporation in pulsation buffer containing 100 µM propidium iodide which is a non-permeant fluorescent intercalating agent. Once plasma membrane presents defaults, propidium iodide penetrates into the cell, intercalates with DNA, and then exhibits a red fluorescence. Cells were trypsinized immediately after electroporation and PI uptake was quantified by flow cytometry (FACSCalibur, Becton Dickinson, San Jose, CA, USA). Cell viability was assessed 24 h after electroporation alone with PrestoBlue reagent (Invitrogen, Loughborough, UK) according to the manufacturer’s instructions. Briefly, 24 h after electroporation, cell culture medium was removed and cells were incubated for 30 min at 37 °C with 100 µl of 1× PrestoBlue reagent diluted in PBS, before reading absorbance on a plate reader at 570 nm and 600 nm (CLARIOstar, BMG Labtech, Ortenberg, Germany). For every set of experiments, three biological replicates were produced and analyzed.

### 4.6. Osmolarity Measurement

The osmolarity of calcium and magnesium solutions was measured using a cryoscopic osmometer, OSMOMAT 030 (Gonotec, Berlin, Germany) according to the manufacturer’s protocol. Four distinct solutions of 5 mM CaCl_2_ or MgCl_2_ in HEPES buffer used along all the experiments were measured. Osmolarity of calcium and magnesium solutions used to treat cells were measured with a cryoscopic osmometer. Solutions of HEPES buffer with 5 mM CaCl2 or MgCl2 presented an osmolarity of respectively 326.3 ± 31.5 and 341.8 ± 12.8 mOsmol/L, which was statistically similar (unpaired t-test, *p* = 0.3973).

### 4.7. Genotoxicity and Cytotoxicity

Genotoxicity and cytotoxicity were determined thanks to the published γH2AX In-Cell Western technique and performed as previously described [19,43,44]. Briefly, HCT-116 tumor cells and normal dermal fibroblasts were seeded 24 h prior to treatment at a respective density of 30,000 and 15,000 cells per well in a 96-well plate. A positive control was added to the plate and consisted of cells incubated for 24 h with 3 µM etoposide (VP16), a known genotoxic drug. Controls conditions were respectively pulsing buffer alone for bleomycin and cisplatin conditions and HEPES buffer for CaCl_2_ and MgCl_2_ conditions. As described above, special attention was paid to the time of contact with the drug (bleomycin, cisplatin, CaCl_2_, or MgCl_2_), so that it was the same with and without electroporation. Then 24 h after treatment, cells were fixed with 4% paraformaldehyde (Electron Microscopy Science) in PBS and chemically permeabilized with 0.2% Triton X-100. Cells were then incubated in blocking solution (MAXblock Blocking Medium supplemented phosphatase inhibitor PHOSSTOP and 0.1 g/L RNAse A) prior to 2 h incubation at room temperature with rabbit monoclonal anti-γH2AX primary antibody (Clone 20E3, Cell Signaling). Detection was carried out with an infrared fluorescent dye conjugated to goat antibody (CF770, Biotium, Hayward, CA, USA). For DNA labeling, RedDot2 (Biotium) was added simultaneously to the secondary antibody. After 1 h of incubation at room temperature, the fluorescence was read using an Odyssey Infrared Imaging Scanner (Li-CorScienceTec, Les Ulis, France). The fluorescence corresponding to γH2AX and co-localizing with RedDot2 was integrated in the constant area corresponding to the one located in-between the electrodes during application of the electric field and expressed as fold change compared with controls in pulsing or HEPES buffer. Cell viability was calculated by relative cell count (final cell count (treated)/final cell count (control) × 100) assessed by automated fluorescence. Genotoxicity was considered positive when the tested treatment produced a statistically significant 1.6-fold γH2AX induction at a level of cytotoxicity above 50% compared to the control. These parameters were based on previous studies based on the use of γH2AX quantification [19,44,45,46,47,48]. All experiments were performed at least three times independently, in duplicate. Error bars represent the mean ± SEM. Statistically significant increases in H2AX phosphorylation and cytotoxicity after treatment were compared between the same drug concentration, associated or not with electroporation using one-way ANOVA followed by Tukey’s post-test (* *p* < 0.05; ** *p* < 0.01).

### 4.8. ATP Quantification

Intracellular ATP content was quantified 5 min after calcium or magnesium electroporation thanks to CellTiter-Glo Luminescent Cell Viability Assay (Promega, Madison, WI, USA) according to the manufacturer’s protocol. This assay uses ATP as a co-factor for luciferase reaction. In brief, 15,000 dermal fibroblasts or 30,000 tumor HCT-116 cells grown in monolayers in 96-well plates for 24 h were incubated 5 min in HEPES buffer containing 5 mM of CaCl_2_ or MgCl_2_, then exposed to electroporation, incubated for 5 min at room temperature, and then washed once with PBS. First, 25 µL of cell culture medium were added and immediately after, 25 µL of reagent were added. The plate was shaken for 2 min on an orbital shaker and then incubated 10 min in the dark at room temperature. Then 40 µL of supernatant were transferred in a white 96-well plate before reading on a luminescence plate reader (CLARIOstar, BMG Labtech, Ortenberg, Germany). Statistical differences between each condition containing six biological replicates were analyzed by one-way ANOVA followed by Tukey’s post-test.

### 4.9. Transmission Electron Microscopy

For an ultrastructural examination of organelles, cells were grown in monolayer on round glass coverslips. Since cells are not supposed to grow on glass, these coverslips were coated with gelatin (Fisher Scientific) chosen because of its high biocompatibility with cells [49]. Then 5 min after calcium or magnesium electroporation, coverslips were plunged in 2% glutaraldehyde in 0.1 M Sorensen phosphate buffer (pH 7.4) and then further processed for transmission electron microscopy. Finally, pictures of ultra-thin sections stained with uranyl acetate were acquired on a Hitachi HT7700 microscope (Hitachi High-Technologies Corp., Tokyo, Japan).

### 4.10. Mitochondrial Membrane Potential

Mitochondrial membrane potential was assessed using MitoView 633 (Biotium) which is a mitochondrial membrane potential-sensitive, fluorogenic dye that rapidly accumulates in mitochondria. Staining is dependent on mitochondrial membrane potential and lost when mitochondria become depolarized. Cells were labeled according to the manufacturer’s protocol. Briefly, cells grown in monolayer were washed with PBS and then incubated for 20 min in the dark in a humidified atmosphere at 37 °C containing 5% CO_2_ with 50 nM MitoView 633 in cell culture medium. Cells were then washed twice in PBS and submitted to calcium electroporation as described above. Then 5 min after treatment, fluorescence pictures (excitation 622 nm/emission 648 nm) of tumor and normal cells were acquired with a wide-field fluorescence DMIRB Leica microscope coupled to a Photometrics Cool SNAP HQ camera.

### 4.11. Quantification of Caspases 9 and 3/7 Activities

Caspases 9 and 3/7 activities were quantified respectively with Caspase-Glo 9 assay and Caspase-Glo 3/7 assay according to the manufacturers’ protocol (Promega). These assays are based on luminogenic substrates of caspases 9 or 3/7 and the signal generated is proportional to the amount of caspase activity present. Briefly, 15,000 dermal fibroblasts or 30,000 tumor HCT-116 cells grown in monolayers in 96-well plates for 24 h were incubated 5 min in HEPES buffer containing 5 mM of CaCl_2_, then exposed to electroporation, washed twice in PBS, covered with 50 µL of cell culture medium, and then placed in a humidified atmosphere at 37 °C containing 5% CO_2_ for the chosen time. After 0, 5, 15, 30, 60, 120 min or 24 h, 50 µL of reagent were added directly into the wells, incubated for 30 min at room temperature and then 90 µL of supernatant were transferred in a white 96-well plate before reading on a luminescence plate reader (CLARIOstar, BMG Labtech). A positive control (cells incubated overnight with 1 µM staurosporine) was added to the experiment to check the quality of the test and cell susceptibility to apoptosis.

### 4.12. Immunofluorescent Detection of Cytochrome c

In order to visualize if cytochrome c was released from mitochondria after calcium electroporation treatment, 15,000 dermal fibroblasts or 30,000 tumor HCT-116 cells were grown in monolayers in 96-well plates for 24 h. They were then incubated for 5 min in HEPES buffer containing 5 mM of CaCl_2_ or MgCl_2_, and exposed to electroporation. Then 5 min after electroporation treatment, cells were washed twice in PBS, and fixed 1 h at room temperature with paraformaldehyde 4%. Cytochrome c was then immunodetected using mouse monoclonal antibody against cytochrome c (Cell Signaling #12963, clone 6H2.B4, dilution 1/300). A positive control for mitochondrial apoptosis leading to cytochrome c release was added (cells incubated 3 h with 50 µM staurosporine).

Three-dimensional (3D) tumor spheroids growth and integrity after 168 mM CaCl_2_ and MgCl_2_ electroporation. Videos presenting the macroscopic aspect of spheroids until 7 days post treatment were obtained on a JuliStage video microscope (NanoEnTek, Waltham, MA, USA). Pictures were taken every 1 h over 7 days. Cell culture media changed at 18 h and 139 h.

## 5. Conclusions

Calcium associated to electroporation allows supra-physiological doses of calcium to enter the cytosol and induces mitochondrial dysfunction leading to cell death. Contrary to what is observed with classical antitumor drugs, such as cisplatin and bleomycin, calcium electroporation does not induce genotoxicity in exposed cells, and therefore turns out to be an inexpensive, efficient, simple, and safe innovative cancer treatment.

## Figures and Tables

**Figure 1 cancers-12-00425-f001:**
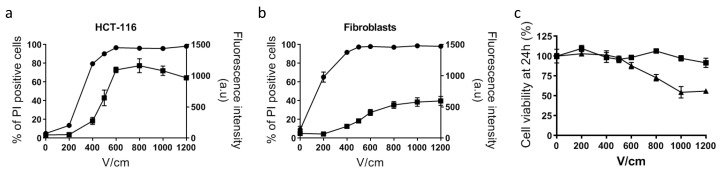
Determination of optimal voltage leading to reversible cell electropermeabilization. Cell electropermeabilization assessed by flow cytometry in HCT-116 tumor cells (**a**) and normal dermal fibroblasts (**b**), *n* = 3. Circles indicate the % of cells labeled with propidium iodide used as a marker of plasma membrane permeabilization. Squares indicate the mean fluorescence intensity (arbitrary units) of positive cells to propidium iodide. (**c**) Cell viability quantified with PrestoBlue assay based on cell metabolism, 24 h after application of electric pulses. Squares: fibroblasts; triangles: HCT-116; *n* = 3.

**Figure 2 cancers-12-00425-f002:**
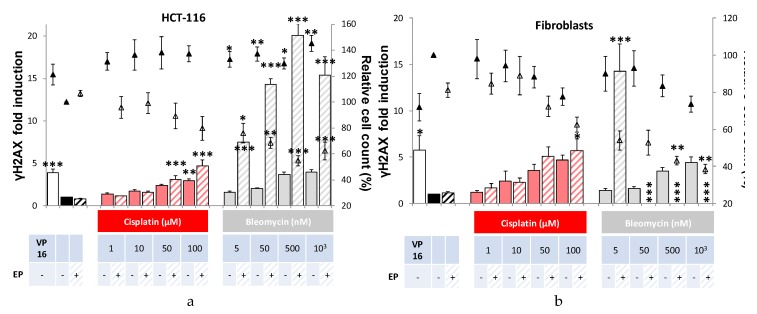
Cytotoxicity and genotoxicity of antitumor drugs bleomycin or cisplatin associated or not with electroporation in HCT-116 tumor cells (**a**) and normal dermal fibroblasts (**b**), 24 h after treatment. Cytotoxicity and genotoxicity of increasing concentrations of antitumor drugs cisplatin (red) and bleomycin (gray) associated or not with electroporation (EP). Induction of γH2AX (histogram) compared with control condition in pulsation buffer. Cytotoxicity associated (hollow triangles) or not (solid triangles) with electroporation is represented by the percentage of relative cell count compared with the control condition (no electroporation). VP16 indicates the positive control (incubation overnight with 3 µM of etoposide). Experiments were performed at least three times independently, in duplicate. Values represent the mean ± SEM. Statistically significant increases in H2AX phosphorylation and cytotoxicity after treatment were compared between each condition and its respective control (i.e., drug alone versus control; or drug associated with electric pulses versus EP) using one-way ANOVA followed by Tukey’s post-test. * *p* < 0.05; ** *p* < 0.01, *** *p* < 0.001.

**Figure 3 cancers-12-00425-f003:**
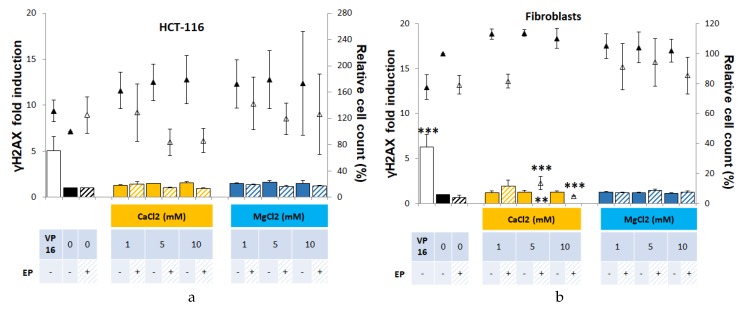
Cytotoxicity and genotoxicity of CaCl_2_ or MgCl_2_ associated or not with electroporation in HCT-116 (**a**) and normal dermal fibroblasts (**b**), 24 h after treatment. Cytotoxicity and genotoxicity of increasing concentrations of CaCl_2_ (yellow) and MgCl_2_ (blue), associated or not with electroporation (EP). Induction of γH2AX (histogram), compared with control condition in HEPES buffer. Cytotoxicity associated (hollow triangles) or not (solid triangles) with electroporation is represented by the percentage of relative cell count compared with the control condition (no electroporation). VP16 indicates the positive control (incubation overnight with 3 µM etoposide). Experiments were performed at least three times independently, in duplicate. Values represent the mean ± SEM. Statistically significant increases in H2AX phosphorylation and cytotoxicity after treatment were compared between each condition and its respective control (i.e., drug alone versus control; or drug associated with electric pulses versus EP) using one-way ANOVA followed by Tukey’s post-test. * *p* < 0.05; ** *p* < 0.01, *** *p* < 0.001.

**Figure 4 cancers-12-00425-f004:**
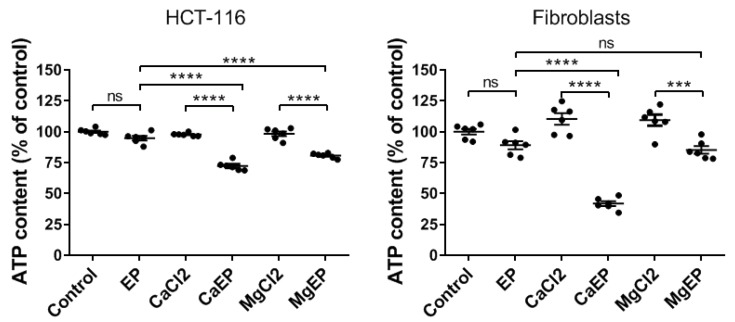
Intracellular ATP depletion within 5 min after electroporation associated with 5 mM calcium. ATP intracellular content was quantified by luminescence assay in HCT-116 cells and dermal fibroblasts. EP: electroporation. Value represents the mean ± SEM (*n* = 6). Statistical differences were analyzed by one-way ANOVA followed by Tukey’s post-test. ns: non-significant. *** *p* ≤ 0.001; **** *p* ≤ 0.0001.

**Figure 5 cancers-12-00425-f005:**
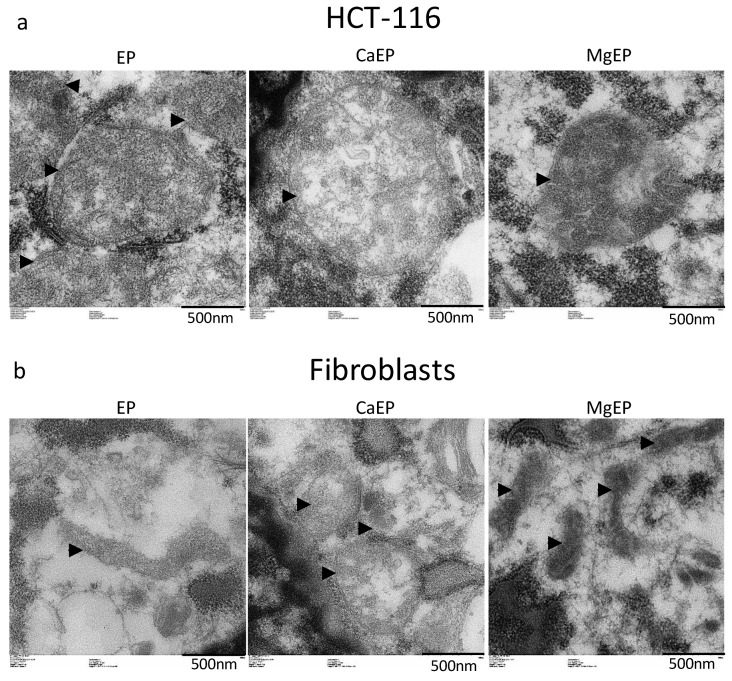
Ultrastructural aspect of mitochondria 5 min post-treatment observed by transmission electron microscopy. (**a**) HCT-116 tumor cells. (**b**) Normal dermal fibroblasts. Arrow heads indicate external membrane of mitochondria. Pictures are representative of what was observed in the cell population.

**Figure 6 cancers-12-00425-f006:**
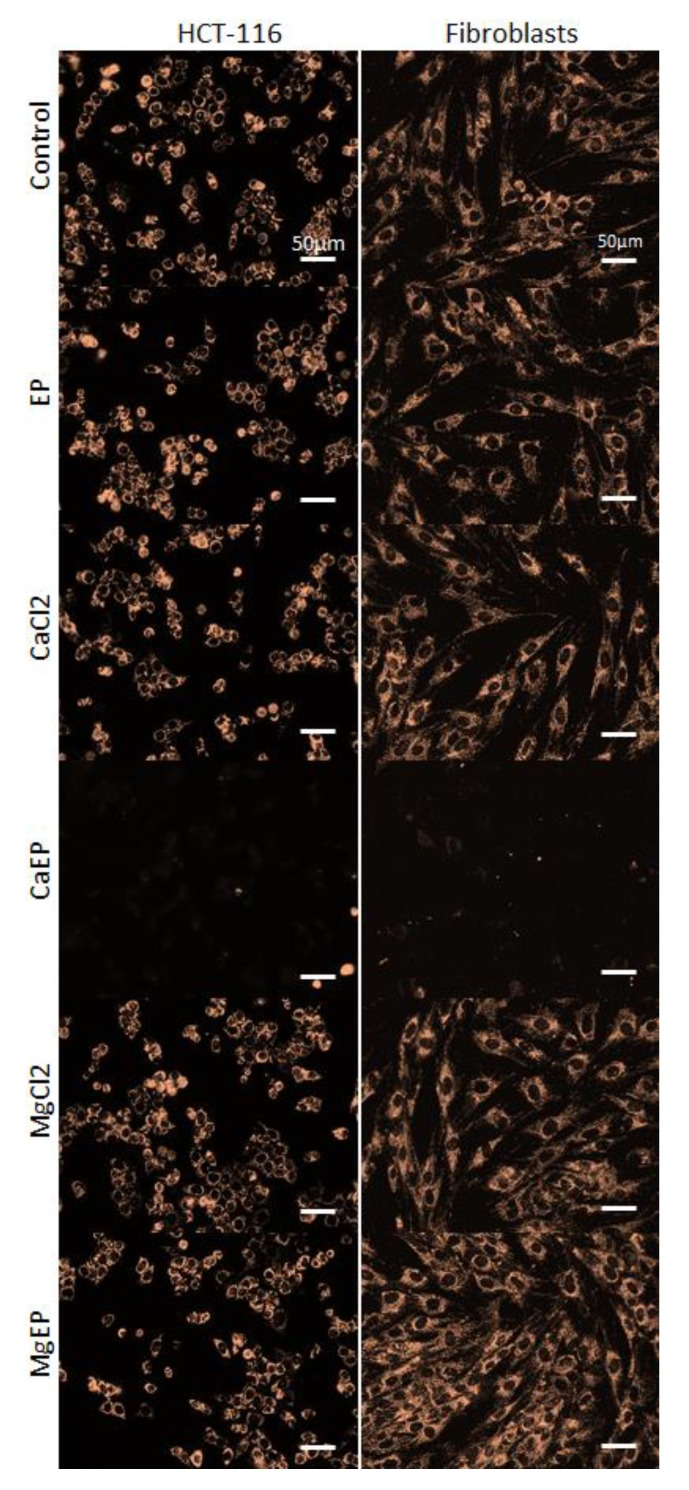
Drastic mitochondrial membrane potential alteration 5 min after electroporation associated with 5 mM calcium. Mitochondrial membrane potential was observed thanks to MitoView633 fluorescent probe in HCT-116 cells and dermal fibroblasts. Representative pictures of two independent experiments.

**Figure 7 cancers-12-00425-f007:**
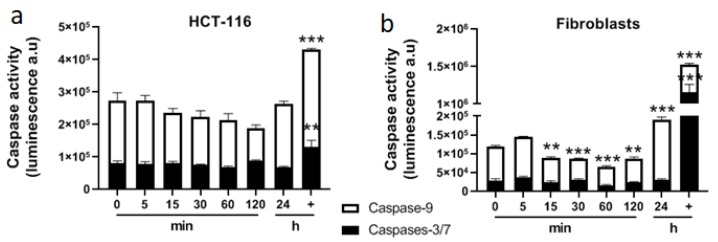
Determination of caspases’ activities after 5 mM calcium electroporation in HCT-116 tumor cells (**a**) and normal dermal fibroblasts (**b**). Caspase-9 and caspases-3/7 were quantified by luminescence. + indicates the positive control (incubation overnight with staurosporine 1 µM). Each value represents the mean ± SEM of at least two independent experiments led in quadruplicate. Statistical differences were analyzed by one-way ANOVA followed by a Dunnett’s post-test in comparison to the control condition (0 min). ** *p* ≤ 0.01; *** *p* ≤ 0.0001.

**Figure 8 cancers-12-00425-f008:**
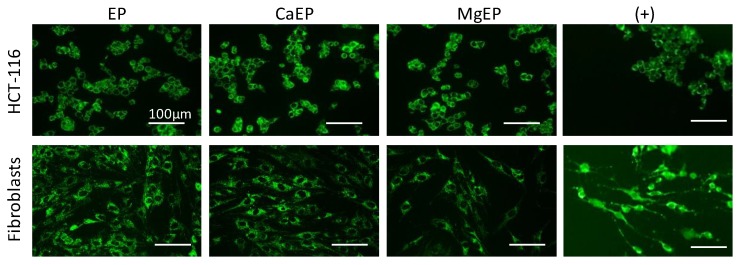
Immunofluorescent detection of cytochrome c 5 min after 5 mM calcium or magnesium electroporation in HCT-116 tumor cells and normal dermal fibroblasts. Punctiform labeling indicates that cytochrome c is sequestered into mitochondria. Indeed, in case of release, the signal is blurry and distributed throughout the cytoplasm, as shown in positive control conditions (+) representing cells incubated for 3 h with 50 µM staurosporine. Pictures are representative of two independent experiments.

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
