# Peer review of "Calcium Delivery by Electroporation Induces In Vitro Cell Death through Mitochondrial Dysfunction without DNA Damages"

_cancers, 2020, doi:10.3390/cancers12020425_

Round 1
Reviewer 1 Report
This was from the outset an excellent paper, bringing completely new evidence about the actual mechanism of action of calcium electroporation. It is likely to become a reference paper for the future.
There were remarks from the reviewers which have now all been adequately addressed.
Thus, the paper should be accepted in the present form.
Reviewer 2 Report
The authors have addressed all my concerns.
But I still cannot see Video S1: long term macroscopic aspect of tumor spheroids to CaCl2 and MgCl2 treatments. After the editor checks the video, I recommend this article for publication.
Reviewer 3 Report
The paper is interesting, technically sound and well organized. The results have been adequately presented and discussed.
This manuscript is a resubmission of an earlier submission. The following is a list of the peer review reports and author responses from that submission.
Round 1
Reviewer 1 Report
The manuscript “Calcium delivery by electroporation induces in vitro cell death through mitochondrial dysfunction without genotoxicity” is a very well written paper with interesting new results within the field. Scientifically it is very sound, methods well described and results and conclusions are in line. There are, however, a few concerns that need to be addressed. Major and minor comments to the manuscript is listed below.
Major:
Although the paper is overall of good, scientific quality with conclusions in line with results there is a section in the introduction that falls decidedly outside the scope of the article – and indeed definitely also outside the field of the authors. Thus, the introduction start with one full paragraph ridden with mistakes about oncological treatments. To exemplify this: “”Cytotoxic drugs, such as radiotherapy” (radiotherapy is not a drug) “Chemotherapy has been used for cancer therapy since the 50’s, and remains the first line 35 treatment for most patients today” (surgery is what cures most cancer patients and is first line treatment). A long description of side effects to chemotherapy does not do justice to the important advances in treating e.g. child leukemia with these drugs. The paper is not at all an oncology paper so this reviewer would insist that the first paragraph (line 35-48) is deleted.
In figure 2 and 3, all symbols are black triangles. It would be easier if EP+ was one symbol e.g. solid triangle and EP- was another symbol e.g. open triangle. It is unfortunately, unclear to the reader what are the controls. Thus, when you write VP16 (μM) across all three bars on the left of the graph in confused the reader to not see that the second bar is untreated control and the third bar is electroporated (no drug) control. Use a symbol instead of a “p”.
The statistical analyses performed should be described thoroughly (type of analysis, how you test for the assumptions, what is done in case of violation of the assumptions etc.) in the Method section. Student´s t-test is not an optimal analysis to perform when you have more than two groups, which you have in the results presented in figures 2 and 3. The authors compare two groups multiple times and thus should use another analysis, e.g. one-way ANOVA to keep Type I error to 5%.
Lines 277-279. As the authors write, different selectivity to calcium electroporation between normal and cancer cells has been observed. However, other studies show no selectivity between normal and cancer cells (see Frandsen and Gehl, Plos One 2017 and Staresinic et al, Scientific Reports 2018). Thus, this sentence should be rewritten or expanded to indicate that a difference in sensitivity has been observed in 3D, in vivo and clinical studies, but (mostly) not in vitro where the external volume of calcium is very large compared to the intracellular volume.
Minor:
When references are cited in groups in the text, references should be listed chronologically in the reference list.
Number of replicates (n) is missing in figures 1, 4, 5 and 8.
Line 86. “calcium treatment” should be “calcium electroporation treatment”.
Line 131. This is result for normal fibroblasts, not HCT-116 cells.
Reviewer 2 Report
In this manuscript, the authors investigated the cytotoxic and genotoxic effects of calcium electroporation using 2D and 3D cell models by comparing the toxic profiles between calcium electroporation, magnesium electroporation and electrochemotherapy. While this paper is of interest to the field, additional experiments and better representations of data should be performed before I can recommend this manuscript for publication.
My specific major concerns are:
On page 3, the authors should show the optimization of electroporation data for 3D tumor spheroids in Supplemental Figures. If already published in reference 12, the same figures should be presented in Supplemental Figures. In Figure 2 the bleomycin panel, the cell viability with EP is between 40-60% in both cell lines. What is the point not showing genotoxicity data when cell viability is below 50%? As the authors mentioned, apoptosis associates with DNA damage, so it is impossible to exclude the genotoxic effects by apoptosis. What is the potency of toxicity of calcium electroporation in 2D tumor cells? The authors should perform a dose curve to show the IC50 values of calcium electroporation treatment to inhibit tumor cell growth. What is the potency of toxicity of calcium electroporation in healthy fibroblasts? Is calcium electroporation more toxic to fibroblasts than tumor cells? In Figure 2 and 3, the authors should clearly mention the cell count (%) is relative to what condition. In Figure 5, it is difficult to see calcium treatment disorganizes mitochondrial inner membranes. The authors should consider using color microscopic picture or increase the resolution to really see the inner membrane clearly. In Figure 6, the authors should present it as color picture since it’s fluorescence microscopy. In Figure 7, the authors should show the caspase activity after 24 hours of electroporation since this is the time when cytotoxicity and genotoxicity is measured (Figure 2 and 3). In Figure 8, a positive control should be added to show the release of cytochrome c since there is no change between the three treatments. The authors really need to show either quantitative results of cell counting or tumor shrinking, or microscopic morphological data for the tumor spheroids in order to support the claims on page 9-10. The movie is missing in the reviewer package.What is the calcium and magnesium concentration in the culture medium?
Some minor points:
The authors should mention briefly in the introduction the in vivo administration mode of calcium electroporation as cancer therapy. Are there any limitations for administration? What’s the scope of calcium electroporation in treating cancer? Is it limited to skin cancer? On page 3, the authors should label the number of replicates in Figure 1.

Reviewer 3 Report
Authors report on an in vitro characterization study of normal and cancer cells subjected to calcium electroporation in order to elucidate the cytotoxic profile of this treatment in comparison with ECT using bleomycin and cisplatin.
The rationale has been clearly stated, the experimental procedures are technically sound and well described. The results have been adequately presented and discussed. I have only some suggestions:
regarding the significance of the increase in γH2AX foci formation. In my understanding, γ-H2AX foci formed per cell nucleus following a treatment reflect the yield of induced double-strand breaks that are highly cytotoxic form of DNA damage which, if not correctly repaired, can initiate genomic instability, chromosome aberrations and mutations and thus genotoxicity [Wyman C, Kanaar R. DNA double-strand break repair: all’s well that ends well. Annu Rev Genet 2006;40:363-83]. For these reasons, with respect to the absence of γH2AX foci induction following calcium electroporation I suggest to refer to absence of DNA damage instead of absence of genotoxicity. regarding the results that both normal dermal fibroblasts and cancer HCT-116 cells exhibit the same effects after calcium electroporation. I suggest to comment on this result. As a matter of fact, several in vitro studies suggest that normal and malignance cells exhibit differential responses to calcium electroporation with a more sensitivity of cancer cells than normal cells. In the caption of Figure 2 (page 4, line 144) “value represents represent the mean…” should be “value represents the mean….”. Statistics for caspases quantification must be described and presented in Figure 7.Author Response
Please see the attachment
